# Static-Analysis-Based Solutions to Security Challenges in Cloud-Native Systems: Systematic Mapping Study

**DOI:** 10.3390/s23041755

**Published:** 2023-02-04

**Authors:** Md Shahidur Rahaman, Agm Islam, Tomas Cerny, Shaun Hutton

**Affiliations:** Department of Computer Science, ECS, Baylor University, Waco, TX 76798, USA

**Keywords:** security, defense, cloud-native, microservice, attacks, vulnerabilities, static analysis

## Abstract

Security is a significant priority for cloud-native systems, regardless of the system size and complexity. Therefore, one must utilize a set of defensive mechanisms or controls to protect the system from exploitation by potential adversaries. There is an expanding amount of research on security issues, including attacks against individual microservices or overall systems and their corresponding defense mechanism options. This study intends to provide a comprehensive overview of currently used defense mechanisms involving static analysis that can detect and react against associated attacks and vulnerabilities. We present a systematic literature review that extracts current approaches for the security analysis of microservices and the violation of security principles. We gathered 1049 relevant publications, of which 50 were selected as primary studies. We are providing practitioners and developers with a structured survey of the existing literature of defensive solutions for microservice architectures and cloud-native systems to aid them in identifying applicable solutions for their systems.

## 1. Introduction

Software developers are adopting several strategies to meet business requirements with the emerging technological shift. The Microservice Architecture (MSA), which fueled cloud-native principles, is now mainstream due the capability it brings to the organization around business essentials. Developers benefit from the high serviceability, loose coupling, and quick deployment of this architectural style [1]. The straight-forward deployment process brings an enormous number of service instances that eases the continuous integration and deployment process [2]. In addition, microservices may be distributed over the network among many execution platforms. By utilizing microservices, large-scale and distributed software systems may be flexible and scalable. However, the distributed nature, massive number of service instances, and substantial inter-service communication through the network bring significant entry points for the potential adversaries [3] to enter the system and potentially inflict harm. Adversaries can exploit those hotspots to violate the security principles of the system.

With significant security challenges, developers must consider security a top priority for the MSA or cloud-native systems. A variety of safeguards are required for the microservices. Each microservice aims to supply functions intrinsically coupled to other microservices through the communication protocol. As a result, if the attackers compromise one of the microservices, they may obtain access to the linked microservices. In addition, it is more challenging to monitor specific services due to their scattered nature. The security challenges go beyond simply locating or blocking the hotspots where the adversaries engage in malicious activity [4]. Instead, we should involve identifying the adversary’s mindset behind their attacks, categorizing it, reacting and responding to the system’s failure, and putting defensive measures in place to mitigate against those vulnerabilities or attacks. Even though researchers have developed several defensive strategies for the MSA, the categorization of these techniques and the identification of potentially malicious behavior still requires attention, given the rise in MSA attacks.

For example, several static analysis tools detect bugs in a monolithic system and techniques utilized as a defensive module. It is also very convenient to perform security analysis for the system. For the MSA, the gigantic structure makes the analyzer work harder to scan for vulnerabilities [5]. Finally, deploying and operating detection and protection mechanisms tend to be tedious work, which indicates that more investigation is needed in this field.

This study conducted a systemic mapping study to uncover the defense mechanisms of cloud-native systems. We comprehensively identified existing research addressing the static analysis of the defensive approach to protect MSA-based systems, categorize potential attacks and vulnerabilities, and provide insight into the current gaps in the existing literature. Finally, with the defenses suggested to mitigate and avoid categorized attacks or vulnerabilities, we used a detailed procedure to extract, categorize, and organize them. To summarize our study, the contributions of the research are:Categorization of the defense mechanism utilizing static analysis addressed toward the microservice and cloud-native architecture.Categorization of the attacks or vulnerabilities and identifying the potential approaches to detect them.A correlational analysis of the defensive approach to preventing the addressed categorized attacks.Summarization of the open challenges in the defensive approach of the MSA resulting from our findings and assembling a statement of intent for the community of scholars and practitioners working in that field.

The organization of this study is as follows: Section 2 describes the “Related Works”. Section 3 follows with our “Research Methodology” reviewing our research method to conduct a systematic literature review, research questions, and inclusion and exclusion criteria. Further, in Section 4, we present the “Research Findings and Results”. Section 5 discusses threats to validity. Finally, we conclude with the “Discussion” and “Conclusions and Future Work”.

## 2. Related Work

The existing secondary study literature provides more general studies concentrating on the specific field of the security aspect of microservices. The specification mainly addresses the threat modeling, authentication authorization, and security mechanism of the MSA. The core aspect of implementing security solutions is identifying the potential challenges and how to address them. Billawa et al. [6] provided security discussions identified in the grey literature and addressed the challenges when implementing security. Finally, the systematic grey literature review identified solutions and best practices, which also analyzed the improvements to existing methodologies in the microservice architectures. Another challenging aspect of implementing a security defense mechanism is the identification of smells indicating potential security vulnerabilities in microservice-based systems. Ponce et al. focused on the well-known smells and their potential mitigations in [7] on 58 primary research works chosen from those published from 2011 through the end of 2020. As a practical benefit for practitioners who may use the findings of their study in their everyday work on protecting microservice-based applications, they finished with an examination of how their methodology molded an important starting point for the security of the MSAs.

Using a systematic literature study, Li et al. [4] thoroughly analyzed evidence-based, cutting-edge quality aspects that were security issues in microservice-based systems. The six most-alarming quality traits (performance, accuracy, scalability, reliability, security, and usability) were determined, along with strategies to address them and their related effects on the systems. Almeida et al. [8] prompted a systematic literature review to address issues pertaining to the difficulties, strategies, and tools for handling authentication and authorization in microservices. Although they focused explicitly on authentication and authorization, their studies vastly related to security mechanisms and the open-source tools and approaches related to that. For example, a rigorous examination of the commercial grey literature on the drawbacks and benefits of designing, creating, and running microservices by Soldani et al. [9] supplemented the academic state-of-the-art. The highlighted pain issues, which also addressed the appropriate granularity of the design of the security policies, were principally caused by the inherent complexity of microservice-based systems. As opposed to this, the benefits are related to the specific characteristics of microservice-based architectures, design patterns that allow for better exploitation, and the ability to deploy and manage the microservices in an application independently. The studies mentioned above primarily focused on security issues, security as quality attributes, and the characterization of security design policies. The categorization of the potential security issues and their corresponding mechanism were uncovered in the above studies. Our study is the first to cover the research on characterizing the defense mechanisms correlating the attacks.

Hannousse et al. [10] developed a systematic mapping that concentrated on the threat categorizations and associated ontologies to address them. They concluded that the threats now being examined and handled by research include those related to the nature, applicable platforms, and validation procedures of security proposals that led to illegal access, sensitive data disclosure, and compromising individual microservices. However, this study focused on the threat categorization not having a generalized category of how can we characterize the potential attacks and how those can be mitigated. To develop secure systems leveraging such security mechanisms, Pereira-Vale et al. [11] detailed the methodology and findings of a systematic mapping analysis to identify the security mechanisms utilized in microservice-based systems published in the literature. In addition, reusing existing architectural expertise to handle security issues in microservice-based systems is made more accessible by the described security solutions. Pereira-Vale et al. [12] also performed a multivocal literature study to extract the current security solutions and introduce categorizations into variants of standard security mechanisms and scopes connected to security contexts. They provided an extensive selection of security measures and procedures. The catalog of security solutions comprehensively provides significant resources for securing microservice architectures. However, it is important to pay attention to how such solutions might categorize and sum up the probable mitigation strategies. Ultimately, these studies define the existing security solutions by addressing current practitioners’ threats, vulnerabilities, and prevention strategies.

According to the combined research topics in Table 1, the aforementioned studies have addressed work on microservices pertaining to the mechanisms and methods of generating systematic mapping studies in microservices describing security problems. The methodologies for examining the defense mechanism of a cloud-native system as it is now implemented have not, however, been covered in any works.

## 3. Research Methodology

This section presents our thorough analysis of the adapted strategies and protocol to employ the systematic mapping study. We start by defining the research questions, searching and retrieving the literature from various data sources for pertinent articles, manually screening the automatically chosen publications to omit those irrelevant to our study, and then, snowballing. Finally, we analyzed these publications to collect statistical and transparent answers to our research questions.

### 3.1. Research Questions

The microservice architecture implementation brought significant security issues addressing several attack hotspots. The risk analysis and the protection mechanism still need attention as the types of attacks and their consequences differ. Practitioners seek potential security solutions when they face any malicious activity. They need to utilize detection methods, which can identify the hotspots considering the architectures of the cloud-native systems, the development strategies, and deployment. Moreover, the detection methods should justify the possible attacks and their corresponding defensive approach. Thus, we aimed to shed light on those strategies and provide awareness of comprehensive static-analysis-based methods to the developers that can be used as an asset to the microservice development and design. Therefore, the research objectives we can adopt are:  

Recognize and categorize attacks and vulnerabilities that affect microservice architectures;Identify the security mechanism utilizing static analysis to defend against those attacks;Identify existing tools or approaches to detect those categorized attacks;Address the identified gaps and focus on the tactics and challenges for each of the objectives above.

With these objectives in mind, we formulated four research questions:RQ1:What are the most-common defense mechanisms for microservices to face security-related issues based on static analysis?(a)What is the taxonomy/categorization of these strategies?RQ2:What attacks and vulnerabilities are addressed by these strategies?(a)What is the taxonomy/categorization of these attacks?RQ3:What tools or approaches exist in the literature?(a)What features do they support?RQ4:What are the current gaps in the defense mechanism based on static analysis?

### 3.2. Searching Procedure

We needed to choose a selection query that was sufficiently broad to collect relevant research articles. To extract the primary research, we looked through five important digital libraries. These are the databases:ACM Digital Library;IEEE Xplorer;Springer Link;Scopus;Science Direct.

The search queries we used to search the above databases are given below:

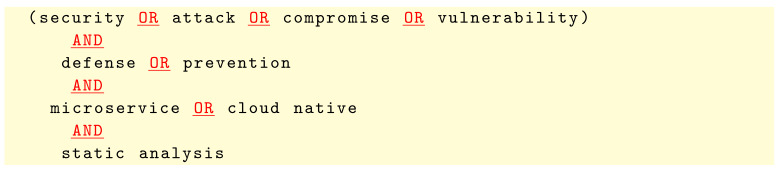



### 3.3. Study Selection

The automated search yielded two filtering phases before the collection of papers. First, the titles and abstracts were read in the initial step to assess relevancy. The second stage involved reviewing the whole texts of the articles to see if they met our inclusion requirements. Then, we used snowballing to find more relevant sources for our research from the works mentioned in the already-chosen articles. Every reference gathered in this manner underwent screening after the first two phases. Finally, the selected articles’ dataset contained all referred papers approved for inclusion by these stages. Until reaching a fixed point or until no new documents were uploaded to the dataset, snowballing was carried out iteratively on these recently added publications.

### 3.4. Inclusion and Exclusion Criteria

Using stringent inclusion and exclusion criteria lowers the number of documents that online academic libraries retrieve. Only peer-reviewed articles from journals and conferences are included in this study. We define those criteria below:Publication published since 2012.Research papers that are in English. Besides being the most-frequently used language by the study team, excluding their native tongue (if applicable), it is also the most prevalent in the technical literature.Publication including studies conducted with the defense of microservices or cloud-native architectures as their primary topics.Papers with full text available in the selected databases.Research papers proposing approaches, frameworks, techniques, methods, or tools to detect attacks or vulnerabilities in microservice or cloud-native using static analysis.Research utilizing the static analysis to detect and address defense mechanisms in microservice or cloud-native architecture.

We similarly define a set of exclusion criteria to comprehensively filter our intended literature. Those are listed below:Research paper addressing strictly network security and protocols in cloud computing.Research papers, not from peer-reviewed sources.Research papers without full text available in the selected databases.Tutorial papers and editorial.Papers describing the general architectural model without mentioning the security aspect in microservices or cloud-native systems.Papers published as short papers (less than three pages).

### 3.5. Data Extraction and Synthesis

We extracted and encoded the relevant data from each primary study after choosing them from the academic literature. At first, we pulled the metadata, which included fields for the name, publication year, source, and type of publications. In addition, we address defense mechanisms with static analysis, corresponding attacks and vulnerabilities, tools, and approaches for solving the issues in the cloud-native system by scanning each selected article. The phases of the filtering process are explained in the link (Search Articles: https://zenodo.org/record/7603720#.Y91UQ3bMK3A), and these procedures are used to filter the article.

### 3.6. Grey Literature

Apart from our academic searching, we also conducted a grey literature review to extract the industrial contributions and practices for security defense mechanisms. An essential addition to a systematic review might come from grey literature or data that have not been published in for-profit journals. In addition, the grey literature may eliminate publishing prejudice, improve review comprehensiveness and timeliness, and encourage a fair picture of existing information. In addition, grey literature materials may be helpful for practitioners and decision-makers across disciplines since they regularly contain updated information that applies to policy and research and is usually simple to obtain. We define several criteria for finding articles, blogs, and journals for the grey literature:We searched Google, StackOverflow, and Quora for relevant studies.Only studies published by professionals with five years or more of experience should be chosen.An analysis must mention at least one industrial case study where a measurable number of microservices are used.Practitioners can significantly obtain the advantages and disadvantages of the issues and topics addressed by the selected studies.Once we found the articles, we categorized them once we replaced the duplicates.

## 4. Results

This section describes the results of the mapping studies and provides a comprehensive answer to the research questions we have defined.

### 4.1. Analysis of the Selected Studies

We conducted the search process in November 2022 and discarded the articles published before 2012. We conducted our article search on the five online databases we utilized for the search, and Table 2 shows the number of articles we found. A total of 1049 research articles were collected initially using our search query. Then, we employed our inclusion and exclusion criteria to filter out 928 articles. Then, we identified inconsistency among the research by scanning the abstracts and titles based on the inclusion and exclusion criteria and removing 49 papers. The remaining 72 articles were then read, and we determined the relevancy considering the research objectives, where we discarded 23 articles. Then, we reduced four duplicate papers, performed the selected studies’ snowballing procedure, and added five documents for consideration. Finally, we considered 50 research documents as our intended focal points of the mapping study. The selection procedure is depicted in Figure 1.

Table 3 shows the in-detail analysis of the selected studies. Figure 2 shows how the selected studies were distributed based on databases and publication year.

### 4.2. Security Defense Mechanism in Cloud-Native Systems: RQ1

Traditional monolithic architectures for creating and delivering programs are being replaced by microservices, revolutionizing systems’ development. The security defense mechanism involves including multi-layer defense against potential attacks and vulnerabilities. Our analysis has categorized the defense mechanisms in cloud-based systems into five categories. The categorization is illustrated in Table 4 and comprises moving target defense, container security issues and their defenses, security-attack-based protections, defense-in-depth, and security-based solutions, which are described next. The distribution is depicted in Figure 3.

Moving target defense: MTD is a strategy that modifies specific system components to make it more difficult for attackers to carry out successful attacks, lowering the attack capabilities. The primary objective is to stop attackers from using information they have learned about target systems due to their homogeneous composition and software monoculture. There were 11.8% primary studies that focused on MTD. Torkura et al. [35] identified the shared vulnerabilities in the microservice architecture and proposed a solution utilizing the moving target defense (MTD), which evaluates the system’s risk assessment. The risk assessment is crucial for the system to identify the potential vulnerabilities, likelihood, and impact. The authors also derived a risk-oriented diversification index and utilized it to alter the attack surface and reduce the attack ability. The potential emergence of security concerns may be unpredictable due to the cloud environment’s modification. To defend the system against attacks, Jin et al. [32] thoroughly analyzed the fixed defensive strategies of MTD. They also provided a framework called DSEOM that could recognize updates to the container-based cloud environment, quickly assess the effectiveness change of MTD, and optimize MTD strategies.Container security issues and their defenses: A lightweight and resource-effective deployment has become easy with Docker container integration in microservice and cloud-native applications. Although we obtain significant advantages from the container, security issues have arisen due to the heterogeneity and unnecessary dependencies and components. We found 29.4% of the studies concentrating on container security issues and their defenses. To safeguard the Docker container against the applications running within inter-service communication and the threats that result from it, the host, and malicious and dishonest hosts, Sultan et al. [25] outlined four primary use scenarios. First, they offered a software-based (which uses Linux kernel capabilities) and hardware-based (which uses trusted platform support and trusted platform modules or TPMs) solutions. Second, Mahajan et al. [41] identified the configuration issues in Docker containers and described potential container deployment security policies that, when properly implemented during the deployment phase of containers, can protect the cloud environment from security intrusions. Shamim et al. [31] offered 11 security practices that aid practitioners in protecting their Kubernetes installations and called for the introduction of RBAC, security patches, and network-specific security policies for the secure deployment of containers. To improve Docker container security based on required access control and to enable container protection without manually configuring it, Zhu et al. [55] evaluated the Linux Security Module and a profile generator named Lic-Sec. Network attacks can adversely influence a cloud system built on containers. By compromising the nodes, the attacker can compromise the internal, external, and virtualization layers (considering a cyber kill chain). As a result, Kong et al. [38] established a Honeynet deployment approach called AHDS to defend against attacks in containers. This strategy uses attack graphs to completely cover and model network attacks in order to successfully defend against attacks in the container-based cloud system.Security-attack-based protections: Researchers have adapted several strategies to protect against several attacks, especially in a cloud-based system. Concerning cross-site scripting (XSS), Bui et al. [17] found vulnerabilities in the APIs of cloud-based applications where external extensions might result in security flaws. For example, adversaries could exploit cloud services’ document sharing and messaging aspects to deliver malicious input. Instead, they provided guidelines, especially with coding practices and security enforcement. To prevent distributed denial of service (DDoS) attacks, Chen et al. [21] focused on developing SplitStack, which can stop attacks by slicing the monolithic stack into several separable components known as minimum splittable units. Li et al. [28] proposed a dynamic DDoS mitigation strategy that can dynamically control the number of container instances serving various users and coordinate the resource allocation for these instances. Finally, the method can maximize service quality to maintain a tolerable service and effectively counter DDoS attacks in the container-based cloud environment. A methodology was put up by Bhargava et al. [48] to ensure that mission-critical cloud systems meet security and performance criteria even in the face of unusual behavior, attacks, and service failure. The approach promises to provide robustness and antifragility under various failures and assaults by actively monitoring the performance and behavior of services. This allows for proactive mitigation of threats and losses in cloud-based systems. The security of the cloud-based system depends critically on intrusion detection and response. To foster autonomy and deep security, Iraqi et al. [50] proposed the Immunizer framework, which uses distributed cluster computing, parallelism, and asynchronous data streaming for monitoring, unsupervised learning for intrusion detection, as well as heuristic-based attack signature generation and intrusion prevention. As a safeguard against potential network attacks, a cost-sensitive adaptive intrusion response system for microservices was proposed by Yarygina et al. [53], which employs a game-theoretic method to respond to network intrusions in real-time automatically. To counter low-level exploitation, Otterstad et al. [60] coupled microservices with software diversity and viewed this as a mitigating method. In comparison to monolithic alternatives, the integration provided higher robustness.Defense-in-depth: Even when using simple-to-manage low-entropy authentication secrets, a flexible communication system with a high level of security may be created by combining standard cryptographic primitives. The method offers encryption, forward secrecy, and protection against replay attacks, even for out-of-order communication. Jander et al. [52] utilized the technique for a better protection scheme. A total of 41.2% primary studies were focused on this category.Framework/Architecture-based Solutions: The management and acquisition of personal health data are secured using the spring framework by Chatterjee et al. [39]. In compliance with the General Data Protection Regulation, the hybrid solution combined the services for sensitive data (TSD) as a service platform and the Hypertext Transfer Protocol security techniques while taking into account security features such as identity brokering, OAuth2, multifactor authentication, and access control to safeguard the microservices architecture APIs. Safaryan et al. [49] created a secure software architecture of microservices that ensures authorized access to confidential data by stating the case for selecting a potential authentication method and creating a security layer that achieves the security objective. The designed architecture enables users to assign or restrict permissions to specific information items using a discretionary or credential approach, protecting them reliably against unwanted access. This category was discussed in 11.8% of the studies we reviewed.

We found two static-analysis-based solutions from our classification that addressed two broad protection aspects. Firstly, the solution mentioned in [39] significantly described the protection for the microservice-based system’s APIs. This solution securely constructed a platform where identity brokering, OAuth2, multifactor authentication, and the access control mechanism ensure proper data safety and user management through API protection. Finally, we can utilize the second solution [49] from our findings for providing confidentially, which also covers the least-privileged principle.

### 4.3. Potential Attacks and Vulnerabilities on Cloud-Native System: RQ2

Potential attacks addressed in cloud-based systems mainly depend on the detection mechanism utilized in the literature. We identified the following attacks/vulnerabilities and their corresponding literature from our analysis. The result and references are depicted in Table 5 and comprise server-oriented attacks, injection vulnerabilities, container attacks, infrastructure and architectural attacks, and denial of service (DoS) attacks. We illustrate the distribution of the category in Figure 4 and detail these next.

Server-oriented attacks: To show how simple it is for an adversary to launch effective attacks on asynchronous web servers that serve several clients concurrently, Morton et al. [33] proposed an instruction tracing approach and a live memory analysis framework. The realism of exploiting memory corruption attacks to compromise these crucial systems was considerably increased by their demonstration of how the control-flow hijacking and privilege escalation phases in the web server exploit chain might be avoided. They also provided several mitigation strategies to counter the attacks.Injection vulnerabilities: Injection happens when a hacker provides the web application with malicious input that is subsequently processed (acted on) unsafely. Thome et al. [36] utilized static analysis and a hybrid constraint to detect injection vulnerabilities in java web applications. They comprehended the attack conditions, then applied satisfiability to that condition to detect the vulnerabilities. Considering the extensive architecture of the cloud-based systems, individual service instances can be prone to injection attacks. This identification scheme can be a good takeaway for the detection scheme for our analysis.Container attacks: In a microservice architecture, we already saw how containers aid the development and deployment procedure efficiently and effectively. However, the system calls of a running container are a potential source of container escalation and brute-force attacks. Zhang et al. [37] provided an intrusion detection system based on a one-class support vector machine (OC-SVM) to detect the attacks we specified. Another intrusion detection system was presented by Almiani et al. [45]. Their detection technique employs a robust backpropagation neural network to identify DDoS attacks in the containerized cloud computing environment. The suggested method sifts through the traffic flows that flowed into the containerized microservices architecture to find malicious DDoS activity. To recognize application-layer CPU-exhaustion DoS attacks in containers, Zhan et al. [43] introduced the coda framework. Coda tracks the amount of CPU time each connection takes and uses statistical techniques to identify attacks. At the host level, it tracks system calls and associated data coming from the Linux-eBPF-based container. When the CPU time used by an attack connection statistically differs from the time used by a genuine connection, an attack can finally be identified.Infrastructure and architectural attacks: Farshteindiker et al. [47] presented an attack vector on the Docker Swarm orchestrator, which is a new concept in offensive security where a cluster is treated as a single unit of processing, an attacker can escalate their privileges in that unit and, after that, perform malicious activity on every component of that unit separately. Alkadi et al. [30] contributed significantly by addressing several attack types, their properties, and their impact on the microservice architectures. The managed attacks are insider intruders, attacks on the hypervisor, flooding, service abuses, port scanning threats, advanced persistent threats, backdoors, and user to root (U2R) attacks. All of the attacks can have a severe impact on the services, e.g., IaaS, PaaS, and SaaS. This study thoroughly analyzed the potential attacks at the architecture level the developers may encounter.Denial of service (DoS) attacks: DoS attacks provide the victim with excessive traffic or information, which causes a breakdown. An approach to efficiently identify application-layer assaults on microservices was put out by Baarzi et al. [51]. Using the Kubernetes Cluster Manager, they constructed their prototype. Performance indicated the viability of the suggested techniques for attack detection and retaliation.Integrity threats in the MSA: Ahmadvand et al. [56] highlighted new integrity issues and conducted a complete security assessment on real-world systems to uncover representative integrity risks involving malicious insiders. Maintaining integrity is one of the security goals. These dangers represent the risk present in such infrastructures, which practitioners may knowingly embrace, reject, or endeavor to reduce.

### 4.4. Tools or Strategies in Security Defense Mechanism: RQ3

From our findings, there are several approaches and tools that the researchers have employed. We provided the references with the addressed approaches and tools in Table 6, and the distribution of the categorization is illustrated in Figure 5. These categorize include security design, container tool, tool considering security principles, detection mechanism, risk analysis, blockchain, and machine learning, and the details are listed below.

Security design: In cloud-based software development, developers should prioritize the security analysis in their development lifecycle. We found that 23.5% of the studies reviewed focused on the security design. Granat et al. [13] provided an identification technique of security approaches in the development lifecycle, which significantly benefits the developers in reducing the security overheads in the software. For microservice development, implementing security architecture tactics impacts secure communication, identity management, and observability. Considering the development stages, architectural design decisions (ADDs) tend to be novel. Zdun et al. [14] introduced techniques to detect those secure tactics of ADDs to provide secure communication, identity management, and observability in microservice systems. Tuma et al. [18] presented a dataset for the security design model, which they utilized to detect security design flaws with five model inspections. Design-level security automation is possible with an acceptable precision value, which provides significant approval of the techniques adopted. Pinconschi et al. [23] developed a tool that automatically repairs software vulnerabilities using the decentralized platform with RESTful APIs and has low overheads. Finally, the tool can be an extensible repair tool for container-based service instances to check and repair vulnerabilities.Container tool: Operating system virtualization is a kind of containerization. Anything from a small microservice or software process to an enormous application might be operated inside a single container. Unnecessary component inclusion needlessly increases the image size, which is potentially vulnerable to security attacks. Rastogi et al. [15] introduced a novel tool called Cimplifier, which can take a container and simple user-defined constraints, partition it into more straightforward containers isolated from each other, and communicate when needed; it has the least resource capability, meaning it utilizes only the resources it needs. By modifying and differentiating the necessary system calls at two different execution phases, namely the booting phase and running phase, Lei et al. [44] presented a container security mechanism called SPEAKER that can significantly reduce the number of system calls available to a given application container. To find the vulnerabilities and create a list of use cases that adheres to the NIST requirements, Ahamed et al. [59] provided a vulnerability-centric method in Docker images. Additionally, they validated the specified use-cases checklist against the OWASP Container Security Verification Standards [61], which businesses may utilize to sharpen the focus of security needs on their projects. There were 17.6% of the studies focused on the container tools.Tool considering security services: The security services that any software system should consider are confidentiality, integrity, and availability. In cloud-based applications ensuring these security services is cumbersome. Clemmys, developed by Trach et al. [20], guarantees users’ functionalities, data confidentiality, and integrity on untrusted cloud premises. We found 5.9% of the studies concentrated on this category.Detection mechanism: If the cloud infrastructure is not established or appropriately constructed, it primarily affects the system’s security. This setup error puts security objectives at risk. For multi-cloud security, Torkura et al. [34] proposed CloudStrike, a security chaos engineering system focused on identifying non-security issues primarily dependent on availability attributes. However, CloudStrike extends the benefits of chaotic engineering to security by introducing security flaws in a cloud infrastructure that affect confidentiality, integrity, and availability. There were 5.9% of the papers centered on this topic.Risk analysis: We already discovered security problems with Kubernetes and container orchestration in general. Helm is a Kubernetes package manager that offers configuration files that specify a programmatic approach for application deployments. Blaise et al. [40] addressed security issues with Helm. They examined those setups, converted Helm Charts into topological graphs, conducted several security studies, and produced a security score and a list of potentially dangerous attack pathways supported by the MITRE ATT&CK framework. This topic was the focus of 5.9% of the publications we discovered.Blockchain: We already defined several approaches comprehensively addressing data security and trust management in [20]. However, including cloud-based blockchain applications is rare in addressing the challenges. Alkadi et al. [30] analyzed the blockchain approach for data privacy and security and how it can collaborate with intrusion detection systems to offer protection and privacy perspectives in cloud systems. In addition, they discussed critical issues with utilizing an IDS and blockchain technology, as well as potential solutions and attack families that would try to take advantage of them. Berardi et al. [46] analyzed and found some of the existing research related to blockchain technology. However, those are more application-based approaches; instead, they are utilized to solve security challenges in the microservice-based system. We observed that 11.8% of the papers reviewed put a focus on this category.Machine Learning: From our analysis, we found that researchers have capitalized on several machine-learning-based approaches. For example, artificial intelligence can strengthen the detection process of attacks and their corresponding classification. Fredj et al. [62] performed similar work by providing several neural network approaches: long short-term memory (LSTM), a recurrent neural network (RNN), and multilayer-perceptron (MLP)-based models to predict the attacks and their behavior. Bhargava et al. [48] presented a model for ensuring mission-critical cloud systems’ security and performance requirements, where service monitoring and mining were performed using unsupervised machine learning. The usage of machine learning mainly focuses on the identification of abnormal behavior to determine a system’s resilience to attacks and failures. Iraqi et al. [50] presented a framework that utilizes autonomic computing and a microagent/microservice architecture approach, and it expands their application-level unsupervised outlier-based intrusion detection and prevention framework. Parallelism, asynchronous data streaming, and distributed cluster computing are also utilized. Almiani et al. [45] introduced the concept of the neural network in cloud-native computing, developing an intelligent network intrusion detection model against the most-contemporary DDoS attacks. Their model successfully identified highly reflective DDoS attacks and can be used to defend against them. Baarzi et al. [51] proposed an unsupervised, non-intrusive, and application-neutral approach to identify application-layer attacks. Additionally, they provided a service fissioning mechanism to isolate the attacker and effectively lessen the impact of the assault on genuine users. For detecting anomalies, Zhang et al. [37] presented an unsupervised method, one-class support vector machine (OC-SVM). Their experimental findings showed that the OC-SVM algorithm could identify contemporary assaults successfully with a good FPR, ranging from 0.02 for brute force attack to 0.12 for adversarial ML attacks. We note that the category was the subject of 29.4% of the studies we collected.

### 4.5. Addressing Current Gaps in the Defense Mechanism: RQ4

Security assessment in a cloud-native system is challenging when employing detection and defensive mechanisms to encounter attacks. We categorized the challenges researchers addressed in the literature into five categories. The categorization and their corresponding references are listed in Table 7, and Figure 6 illustrates the categorization’s distribution, which comprise challenges related to container, edge and fog computing, systematic literature reviews, practitioners, and system design, and the details are given below.

Challenges related to containers: Watada et al. [27] presented several prospective containerization concepts, including technical details on security and isolation, efficient administration, and orchestration necessary for their successful industrial implementation. They also gave a thorough understanding of various research issues and possible directions for lightweight virtualization. Finally, they offered recommendations on how containerization should proceed to overcome the difficulties. The container’s debloating concept was introduced by Rastogi et al. [33] in the earlier section of the defensive approach. However, the primary concern is that, if the analysis is incomplete, it might not detect all the necessary resources. Thus, to provide a better security aspect in the container, Rastogi et al. [16] updated the concept with two new considerations where both dynamic and static analysis were integrated and test case augmentation using symbolic execution. Manu et al. [29] governed container security using the cloud’s platform as a service (PAAS) protection. They compared the safety of virtual machines with and without hypervisors and container technologies. They finally gave some thought to the suggestions for achieving a multilateral balanced security solution for Docker containers by consistently applying hardening security methods. We saw that 26.7% of the studies were devoted to this category.Challenges related to edge and fog computing: Both concepts extend cloud computing and provide better security in the cloud environment. Caprolu et al.’s [24] analysis of virtualization technologies’ effects on the edge/fog network architecture included their benefits and the security concerns they raised. They also updated the prevalent security challenges in such designs and offered some options for future research that might impact general security. Yu et al. [57] assessed the various security concerns that microservice-based fog applications encounter. They focused on the containers, data, permissions, and network components of microservices’ communication of services. Finally, they offered a solution to address network attacks and software-defined network (SDN) security vulnerabilities to close the existing security gaps. Notably, 13.3% of the papers were devoted to this topic.Systematic literature review: Existing mapping studies are good resources when analyzing core concepts, as they provide several in-depth challenges, benchmarks, and guidelines. We obtained several systematic mapping studies from our findings concerning security mechanisms, security threat detection, and mitigations. These studies provide some thorough analysis, reducing the research gaps in this field. Pereira-Vale [11] conducted a mapping study of 26 articles to identify the security mechanism used in microservice-based systems. Their findings can be an excellent resource to add to the existing architectural knowledge to address security problems in microservices-based systems. From the perspectives of the threat model and mitigation, Berardi et al. [46] described security as being at an early stage and addressed critical security attacks involving microservice architectures. They covered the connection between the primary microservices’ development methodologies and security and modern infrastructure security solutions. Finally, Hannousse et al. [10] categorized the security threats and mechanisms. In contrast to prevention and mitigation, auditing and imposing access control are the strategies that have received the most-significant research attention, according to their study. It is noteworthy that the subject made up 20% of the research reviewed.Challenges for practitioners: For software developers, it is challenging to consider the security assessment for secure development and deployment. Weir et al. [19] comprehensively analyzed how a software development team intervention might enhance security. Additionally, they explained the significance of engineers being able to portray security improvements in terms of their commercial benefits. This analysis is necessary for the practitioners to assess and evaluate systems in the development stage to encounter security issues in software. Yarygina et al. [26] focused on how the microservice architecture affects security. Their contribution improves the analysis while tackling the difficulties of merging distributed systems, service orientation, and the basic principles of software engineering. In addition, they provided security recommendations along with a straightforward security architecture for microservices that practitioners may use. The knowledge gap on effectively protecting a microservices system among practitioners was highlighted by Ali et al. [54]. They concluded that the 28 practices were beneficial, as indicated by the survey respondents. Eventually, their list of best practices for microservices security can be a valuable tool for practitioners to handle security concerns in microservices systems. We discovered that this category was the focus of 20% of the primary investigations.Challenges in system design: Due to an elevated attack surface and an excessive cognitive burden for security analysts, the enormous structure of microservices raises serious security concerns. For the architectural design of microservice applications, Tukaram et al. [22] inventoried several pertinent security rules and assessed how these rules may be verified automatically. In addition, their standards gave substantial instructions for the secure configuration of the deployment infrastructure for microservice applications. Flora [42] solved issues with settings that have microservice architectures, multi-tenancy, heterogeneity, and systems dynamicity. He advised that container environments are suitable for host-based intrusion detection. This will be expanded to include dynamic situations as a first step in studying intrusion tolerance strategies appropriate for multiple configurations. In cloud-native architecture, it is crucial to have a central point where the security policies can be implemented. Torkura et al. [58] addressed design patterns that introduce security issues and introduced the idea of a security gateway to address the challenges. This gateway serves as a practical security enforcement point for enforcing security policies, such as ensuring that microservices are pushed into production and do not have any vulnerabilities mentioned explicitly in the security policy. We observed that this category received 20% of the attention in the primary research.

### 4.6. Result Extraction of Grey Literature

Industry practices can provide a comprehensive overview of how developers and security analysts encounter security issues, what approaches they practice, and how they mitigate those while continuously delivering the product to the customer. Organizations and communities comprehensively construct security acts, policies, and schemes to protect against threats and attacks [63]. This implementation can provide an in-depth overview to safeguard data and ensure security goals for the organization’s stakeholders. From our result analysis, we considered the top 50 result studies from our grey literature search. After extraction, we grouped those findings into several categories. Then, we identified and removed the duplicates to make the result consistent. We considered a total of 16 results for the conclusion, which are listed below in Table 8. From our extraction, illustrated in Figure 7, we can see that 43.8% of the selected studies primarily focused on the tools and approaches, 25% of the studies concentrated on best practices and security guidance, and the rest were articles focusing on addressing threats.

Tools and approaches in industry: For cloud-native security, we investigated the article [64], where the four Cs of the cloud: cloud, container, cluster, and code, were analyzed. It also suggested five strategies that comprehensively ensure the security of cloud-native applications: shared dependencies, shifting left (which focuses on applying vulnerability scans in the early development process), managing vulnerable dependency packages, implementing defense-in-depth mechanisms, ad employing a cloud-agnostic security platform. The article [65,67] similarly analyzed the four Cs of cloud security. They mentioned those categories’ common issues and how they can be addressed. A distributed, adaptable, and responsive zero-trust security paradigm from IONATE [66] took the data input, source, type, and gateway into account. This model detects anomalies and mitigates them while monitoring and learning from the interactions between these numerous components. For DevOps and DevSecOps, CloudStrike [68] provides a security solution tool that controls the application lifecycle, encompassing workloads, containers, security posture, and compliance. It offers visibility and security for private, public, hybrid, or multi-cloud settings. Finally, the task and scanning process automation enhances productivity and reaction times while preventing dangers. In addition to the advantages the cloud-native application protection platform (CNAPP) provides, a risk-based solution was presented in [69]. This tool, named Apiiro, is a rapid, context-sensitive static analysis and NLP engine that addresses critical risks such as design flaws, configuration errors, architecture drifts, trade secrets, and supply chain attacks in the application code. Xenonstack [70] discussed the guidance of DevSecOps, which implements security at every step in the DevOps Lifecycle with DevSecOps Tools. They discussed the DevSecOps Tools: Continuum Security, Checkmarx, GauntIt, etc., including the DevOps Pipeline. Similarly, they comprehend several tools: IriusRisk, ThreatModeler, and OWASP Threat Dragon.Security guidance: Snyk [71] provided a comprehensive comparative analysis between cloud-native security and legacy tools. They provided security guidance for cloud-native applications. Their infrastructure-level security guidance can be a good takeaway for practitioners to ensure security in cloud-native environments. Renowned Security Expert Chris Wysopal discussed the evolution of safety in cloud-native applications in Veracode [72]. He mentioned several aspects of application security, how CNAPP works, and why CNAPOP is better. He provided the idea of how application security is becoming complex in the industry rather than performing some simple execution. Synopsys [73] discussed some of the sub-domains of cybersecurity where cloud safety has been analyzed. The analysis provides the typical cyber attacks and comprehends the difference between breaches and attacks. In addition, the best practices were suggested to provide in-depth analysis to protect against security breaches. TechBeacon’s analyst [74] provided an overview of the importance of application security. He mentioned static and dynamic analysis security tools and how those can be utilized. He suggested how security teams can ensure protection for open-source or even legacy apps. The best-recommended application security tools and a view to employing a maturity model for them were presented.Best practices for cloud-native applications: The best practices can guide software developers and security analysts to provide protection against security attacks and breaches for the baseline. Styra’s [75] software analyst provides seven fundamental best practices for cloud-native applications. The first practice is shift left, which we already covered in the tools and approaches. The second practice specifies the security by design that considers dynamic analysis security testing (DAST) and static analysis security testing (SAST). In the third practice, he mentioned defense in depth. For authentication and authorization, he suggested utilizing SAML, WS-Fed, or OpenID Connect/OAuth2 standards-based identity and access management (IAM) for user authentication. The best practices addressed were implementing an API gateway, ensuring the container’s security, and finally, service-to-service communication. SecurityCompass [76] analyzed the security challenges when we havethreat models in microservice-based applications. The eight best practices it presented were adopting a security culture, encryption, issuing and expiring credentials quickly, decoupling security policies, and decoupling security policies introduced newly in our analysis. Hackerone [77] described the threat modeling concept and introduced some application security tools. Finally, the four best practices were presented, where asset tracking and managing privileges are newly adopted practices in consideration. Benison’s [78] security guidelines produce eight best practices where several best practices have already been addressed in our analysis. Access control, user identification, and automatic security upgrades are brand-new ideas for an economical and flexible way to offer practical applications.Threat Analysis: The evolution of security threats in microservice architectures was analyzed in a research work that we found in [79]. The thesis stated that threat modeling using attack graphs and attack simulations was used to examine dangers in microservice architectures and how they relate to design patterns. In addition, a meta-attack language has been used to formalize the attack graphs in two different experimental analyses.

## 5. Threats to Validity

We explore the threats to the validity of our study in this part, including constraints on construct validity, external validity, and internal validity.

Construct validity: The construct validity of our study focuses on the operational measures that are studied to represent the goal and how we investigated while considering the research questions. The identification of primary research from the papers available in the literature is also reflected in it. The core aspect of designing the search strategy was the research topic, which guided us thorough the selection of the search query. Finally, we made two iterations of snowballing to feature additional research for consideration. Finally, we devised a set of stringent inclusion and exclusion criteria to ensure the inclusion of excellent papers, where only peer-reviewed journal and conference papers were accepted for their completeness and adequate findings.Internal validity: We conducted a careful approach to maintain our findings’ internal validity. The collection of data from the selected studies poses a danger to internal validity. To lessen the risks, we developed a technique that involved searching for relevant literature using the specified keywords and then applying a snowballing process backward to the chosen papers. The grey literature was also left out.External threats: The study’s generalization of its findings threatens external validity. The categorization systems used in this mapping were created using the literature obtained. Whereas many others might not, future published research might match the suggested framework. The mind map construction concentrated on the challenges and future extensibility of the investigation.

Threats to the validity of the results focus on problems that limit the capacity to make the correct conclusions. For example, we adhered to the recommended best practices by Petersen et al. [80] to reduce any risks to the result’s validity.

## 6. Discussion

The mapping study guided us in establishing six core concepts of microservice security analysis. Considering the concepts, we constructed an initial mind map to review microservice security. We already discussed the potential defense mechanism in the Results Section. The mind map is illustrated in Figure 8.

Security attacks: Considering the architecture and implementation of a cloud-native system, we found five general attacks that can occur. DDoS attacks are prevalent, where CSRF and XSS can arise considering the web implementation of the systems. Considering the system’s distributed nature, the database can also be prone to injection attacks. Owing to the communication between the service instances and the client request and response, replay attacks might occur to circumvent network communications.Security flaws: The security flaws in cloud-native architectures we encountered are categorized into three broad categories: container-based, deployment, and design. The significant literature demonstrated the issues related to service Docker containers and their corresponding flaws. The architectural flaws and lack of security assessment in the software development phase can significantly impact the system’s security goal. Finally, deployment flaws, which are a critical aspect of cloud-native systems can create security issues considering the scalability and fast delivery.Detection mechanism: To detect the attacks, developers must implement the potential detection mechanisms. The request–response mechanism should be checked using distributed tracing, where we can integrate the pen-testing to evaluate the security. In addition, we can implement a web application firewall for the web servers to evaluate the traffic and communication for the information flow. We can also evaluate the system using several scanning applications such as ZAP to detect attacks or vulnerabilities. Finally, static analyzer tools also provide significant benefits in the detection strategy.Security strategies: To ensure proper security, we must consider some core aspects that provide the security goals in cloud-native systems. Proper resource management can give the system availability, which is one of the primary security goals. Authentication and authorization must justify the access control for the system that provides integrity. Finally, data confidentially should be maintained using the access control mechanism. Secure communication should follow the appropriate traffic management, e.g., API gateway, Transport Layer Security (TLS), token-based secure communication, certificate-based communication, and key control.Industry practices: We found several industry best practices. Some of the best practices have been drawn by academic researchers, and some still need attention. The following will discuss some industry best practices we can utilize while implementing static security solutions against security breaches.Implementing OAuth for identity and access control: OAuth can handle token-based access control, which can handle identity management for the organization. Utilizing it can allow developers to leverage libraries and platforms that significantly speed up development. In addition, some of the most-prominent organizations and most-knowledgeable engineers have already developed several methods for enhancing the security level of your OAuth-based authorization service.Least privilege: A microservice should only be granted the access rights necessary to perform its function. Each microservice and system component should only receive the absolute minimum amount of permission.Automatic security updates: It is convenient to find a means to automate or at least keep the security upgrades under control in the early stages of development if we want our microservices architecture to be safe and scalable simultaneously.Implementing firewall on API Gateway: One method for efficiently handling several service interfaces is an API gateway. Inside the microservices architecture, this tactic can allow some firewall security. This can effectively surround all the microservices with a firewall by hiding the API gateway behind one. For example, the attack surface may be protected with a scalable layer by properly handling permission and authentication.Using scanner for containers: Docker local image vulnerability scanning enables developers to assess the security issues of the container images and take corrective action to address vulnerabilities discovered during the scan, leading to more secure deployments.

Apart from the practices we mentioned above, we already assessed the security issues and addressed them with the initial mind map we have developed, which can significantly guide the practitioners. This mind map will need to be iteratively expanded over time, but it serves as a starting point to guide practitioners.

Additional consideration: We already discussed several detection mechanisms of threats and attacks in microservice architectures. Recent technological advancements guided more sophisticated approaches that comprehensively detect anomalies in data privacy and system security. In addition, the development of microservices is susceptible to the formation of abnormal system behaviors for several reasons, including their deployment in the network and the usage of various technologies. Similar to the previous point, it is challenging to effectively monitor the security and behavior of microservice settings due to the high complexity of small services.Blockchain: As a means of establishing credibility, integrity, and consistency across these interconnected systems, blockchain technology is an excellent tool for coordinating data, or state information, across services. A significant problem is coordinating user requests across time since microservices operate in dynamic contexts that produce concurrent inquiries. The blockchain serves as a central repository for agreed-upon truth since it is an immutable, append-only ledger that may be locked at a particular moment. This enables coordination and reliability across several microservices, regardless of how complicated or dynamic they may be. With blockchain technology, it is possible to protect permissions with the data at the source, allowing confined services to skip the discovery layer and ensure they never access or provide clients with data that were not authorized.Employing machine learning: To make quick and precise predictions and judgments, machine learning has demonstrated its effectiveness in evaluating data acquired and stored in cloud infrastructures. In addition, practitioners utilize AI to discover, research, and anticipate possible external attacks and vulnerabilities in microservice-based ecosystems, enhancing their security. For intrusion detection, unsupervised learning methods can be used. The detection mechanism can be made easy with the integration of a machine learning approach, and we already acknowledged the implementation of neural networks for identifying and preventing DDoS attacks and application-layer attacks. Furthermore, decentralized learning systems offered by AI-based solutions can improve security in microservice-based applications by tracking the data flow required to maintain coordination between the containers containing these entities.

## 7. Conclusions and Future Work

This article comprehensively presented a systematic mapping study of defensive mechanisms using static analysis in cloud-native systems. The study investigated 50 research articles from 1049 utilizing inclusion, exclusion criteria, and snowballing. The result demonstrated that the existing approaches are concentrated on container vulnerabilities and their security implementations. In addition, security design flaws, and detection mechanisms convincingly addressed several attacks: DDoS, CSRF, SQL Injection, XSS, and Replay. Finally, we constructed an overview that provides a significant analysis for the practitioners on how they can utilize the studies, tools, and suggestions which potentially guide us to address the need to remove the research gap in the existing literature. Eventually, with the progressing security issues, we will develop a static analyzer that will implement a security defense solution addressing potential adversarial categorized attacks, detecting them, and mitigating them in our future study.

## Figures and Tables

**Figure 1 sensors-23-01755-f001:**
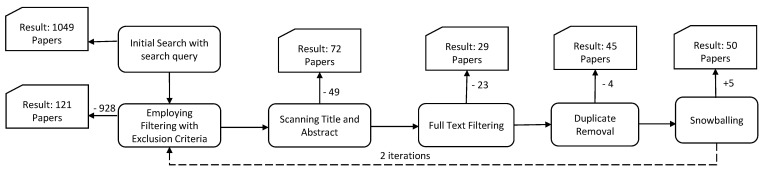
Selection procedure.

**Figure 2 sensors-23-01755-f002:**
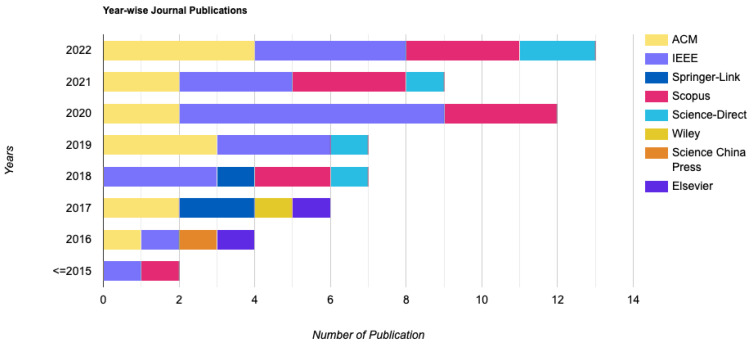
Selected studies distribution by year.

**Figure 3 sensors-23-01755-f003:**
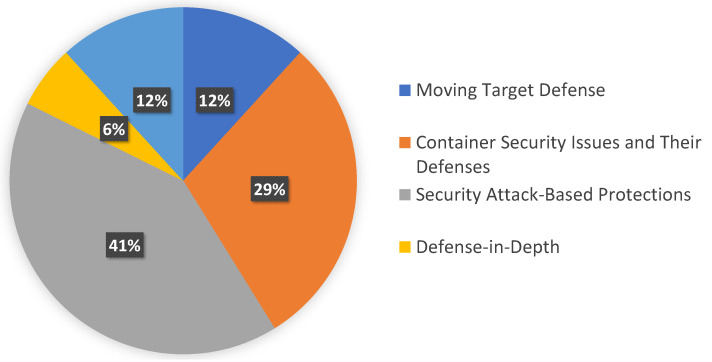
Distribution of categories for defense mechanisms.

**Figure 4 sensors-23-01755-f004:**
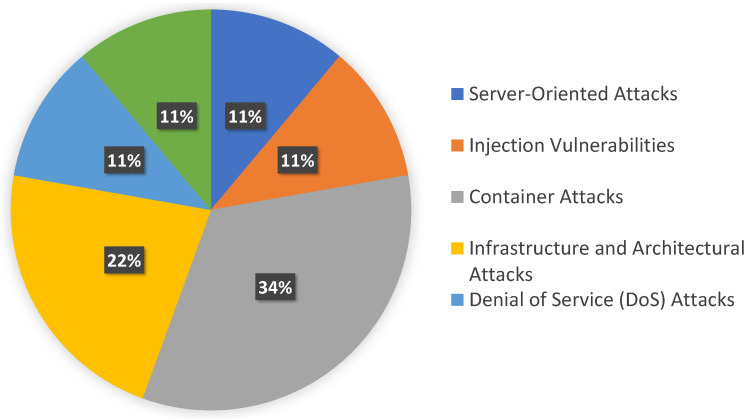
Distribution of categories for attacks.

**Figure 5 sensors-23-01755-f005:**
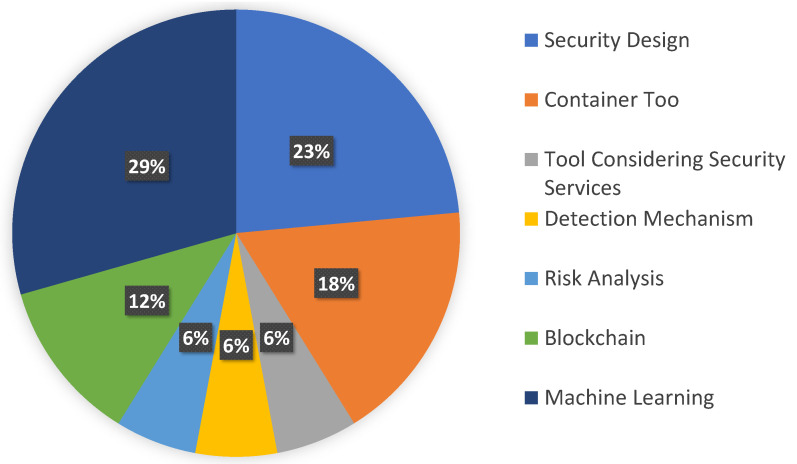
Distribution of categories for tools or strategies.

**Figure 6 sensors-23-01755-f006:**
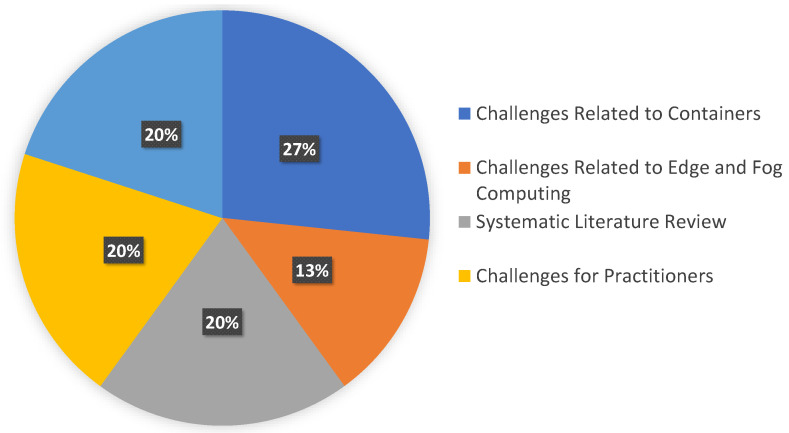
Distribution of categories for current gaps.

**Figure 7 sensors-23-01755-f007:**
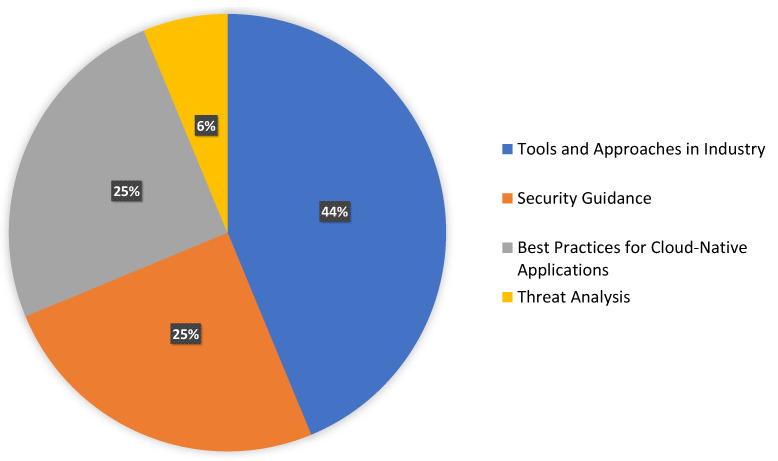
Distribution of categories for industry practices.

**Figure 8 sensors-23-01755-f008:**
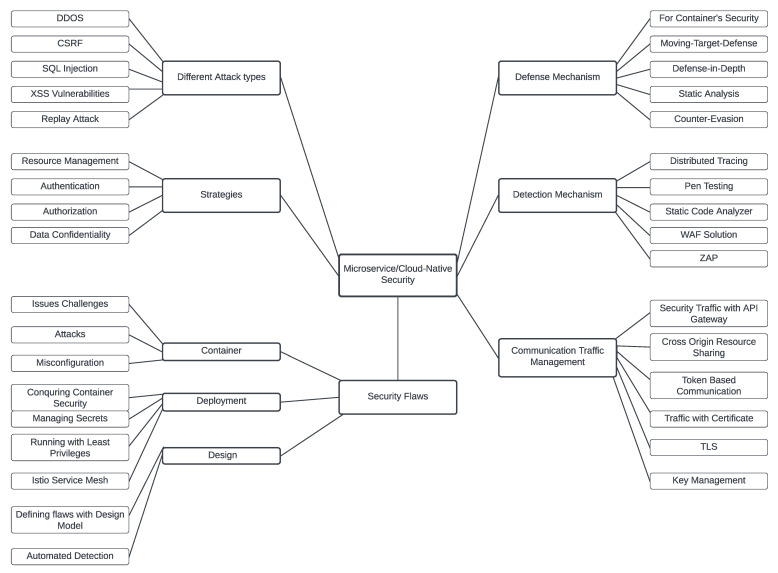
Categorized security research topics related to static analysis in cloud-native systems.

**Table 1 sensors-23-01755-t001:** Research questions of existing secondary studies in related work.

Research Questions Addressed in Related Works	
**Research Question**	**Citation**
What are the current challenges reported by practitioners in the field of microservice security? How do practitioners address the challenges mentioned in RQ1 and what are their recommendations to overcome these challenges? (a) What best practices are mentioned by practitioners? (b) What technical solutions do practitioners propose?	[6]
Are there well-recognized smells indicating possible security violations in microservice-based applications? How can microservice-based applications be refactored to mitigate the effects of security smells therein?	[7]
What are the most-concerned QAs for the MSA? What tactics have been proposed or discussed to improve the most-concerning QAs of the MSA?	[4]
What are the challenges mentioned in the literature to perform authentication and authorization in the context of microservice architecture systems? What mechanisms are used in the literature to deal with the challenges related to authentication and authorization in a microservices architecture? What are the main open-source technology solutions that implement the authentication and authorization mechanisms identified in the literature?	[8]
How much evidence of microservices experimentation from the industry is available online? What are the technical and operational “pains” of microservices? What are the technical and operational “gains” of microservices?	[9]
What are the most-addressed security threats, risks, and vulnerabilities of microservices and microservice architectures, and how can they be categorized? What are existing approaches and techniques used for securing microservices and microservice architectures, and how can they be categorized? At what level of architecture are the proposed techniques and approaches applicable for securing microservices? What domains or platforms are the focus of existing solutions for securing microservices and microservice architectures? What kind of evidence is given regarding the evaluation and validation of the proposed approaches and techniques for securing microservices and microservice architectures?	[10]
Which security mechanisms have been reported in microservice-based systems research? (a) In which security categories do these mechanisms fall? Which empirical strategies have been used to validate research on security mechanisms? Which research strategies are used in the research of security mechanisms for microservice-based systems?	[11]
How has the frequency of publications on security in microservice-based systems varied over time? How have the selected publication publishers changed? Security Solutions’ Categorization (a) What security mechanisms have been proposed or studied in microservice-based systems? (b) What is the security scope of studies in microservice-based systems? What security contexts have been addressed by the research?	[12]

**Table 2 sensors-23-01755-t002:** Papers extracted from particular digital libraries.

Documents by Each Database	
**Database**	**Results**
ACM	121
IEEE	214
Scopus	687
SpringerLink	11
Science Direct	16

**Table 3 sensors-23-01755-t003:** Extracted and analyzed primary studies.

Articles Selected from Journals				
**Article ID**	**Article Name**	**Year**	**Journal**	**Cite**
A1	MetaSEnD: A Security Enabled Development Life Cycle Meta-Model	2022	*ACM*	[13]
A2	Microservice Security Metrics for Secure Communication, Identity Management, and Observability	2022	*ACM*	[14]
A3	Cimplifier: Automatically debloating containers	2017	*ACM*	[15]
A4	New Directions for Container Debloating	2017	*ACM*	[16]
A5	XSS Vulnerabilities in Cloud-Application Add-Ons	2020	*ACM*	[17]
A6	Automating the early detection of security design flaws	2020	*ACM*	[18]
A7	A passion for security: intervening to help software developers	2020	*ACM*	[19]
A8	Clemmys: towards secure remote execution in FaaS	2019	*ACM*	[20]
A9	Dispersing Asymmetric DDoS Attacks with SplitStack	2016	*ACM*	[21]
A10	Towards a Security Benchmark for the Architectural Design of Microservice Applications	2022	*ACM*	[22]
A11	Maestro: a platform for benchmarking automatic program repair tools on software vulnerabilities	2022	*ACM*	[23]
A12	Edge Computing Perspectives: Architectures, Technologies, and Open Security Issues	2019	*ACM*	[24]
A13	Containers’ Security: Issues, Challenges, and Road Ahead	2019	*IEEE*	[25]
A14	Overcoming Security Challenges in Microservice Architectures	2018	*IEEE*	[26]
A15	Emerging Trends, Techniques and Open Issues of Containerization: A Review	2019	*IEEE*	[27]
A16	Exploring New Opportunities to Defeat Low-Rate DDoS Attack in Container-Based Cloud Environment	2020	*IEEE*	[28]
A17	A study, analysis and deep dive on cloud PAAS security in terms of Docker container security	2016	*IEEE*	[29]
A18	A Review of Intrusion Detection and Blockchain Applications in the Cloud: Approaches, Challenges and Solutions	2020	*IEEE*	[30]
A19	XI Commandments of Kubernetes Security: A Systematization of Knowledge Related to Kubernetes Security Practices	2020	*IEEE*	[31]
A20	DSEOM: A Framework for Dynamic Security Evaluation and Optimization of MTD in Container-Based Cloud	2021	*IEEE*	[32]
A21	Security Risks in Asynchronous Web Servers: When Performance Optimizations Amplify the Impact of Data-Oriented Attacks	2018	IEEE	[33]
A22	CloudStrike: Chaos Engineering for Security and Resiliency in Cloud Infrastructure	2020	*IEEE*	[34]
A23	Security Mechanisms Used in Microservices-Based Systems: A Systematic Mapping	2019	*IEEE*	[11]
A24	A Cyber Risk Based Moving Target Defense Mechanism for Microservice Architectures	2018	*IEEE*	[35]
A25	An Integrated Approach for Effective Injection Vulnerability Analysis of Web Applications Through Security Slicing and Hybrid Constraint Solving	2020	*IEEE*	[36]
A26	A real-time intrusion detection system based on OC-SVM for containerized applications	2021	*IEEE*	[37]
A27	Automated Honeynet Deployment Strategy for Active Defense in Container-based Cloud	2020	*IEEE*	[38]
A28	SFTSDH: Applying Spring Security Framework With TSD-Based OAuth2 to Protect Microservice Architecture APIs	2022	*IEEE*	[39]
A29	Stay at the Helm: Secure Kubernetes deployments via graph generation and attack reconstruction	2022	*IEEE*	[40]
A30	Detection, Analysis and Countermeasures for Container based Misconfiguration using Docker and Kubernetes	2022	*IEEE*	[41]
A31	Improving the Security of Microservice Systems by Detecting and Tolerating Intrusions	2020	*IEEE*	[42]
A32	Coda: Runtime Detection of Application-Layer CPU-Exhaustion DoS Attacks in Containers	2018	*IEEE*	[43]
A33	SPEAKER: Split-Phase Execution of Application Containers	2022	*Springer-Link*	[44]
A34	Resilient Back Propagation Neural Network Security Model For Containerized Cloud Computing	2022	*Scopus*	[45]
A35	Microservice security: A systematic literature review	2018	*Scopus*	[46]
A36	Securing microservices and microservice architectures: A systematic mapping study	2021	*Scopus*	[10]
A37	Leadership hijacking in Docker swarm and its consequences	2021	Scopus	[47]
A38	Privacy-preserving data sharing and adaptable service compositions in mission-critical clouds	2021	*Scopus*	[48]
A39	Information system development for restricting access to software tool built on microservice architecture	2020	*Scopus*	[49]
A40	Immunizer: A Scalable Loosely-Coupled Self-Protecting Software Framework using Adaptive Microagents and Parallelized Microservices	2020	*Scopus*	[50]
A41	Microservices made attack-resilient using unsupervised service fissioning	2020	*Scopus*	[51]
A42	Defense-in-depth and Role Authentication for Microservice Systems	2018	*Scopus*	[52]
A43	A game of microservices: Automated intrusion response	2018	Scopus	[53]
A44	An empirical study of security practices for microservices systems	2022	*Science-Direct*	[54]
A45	Lic-Sec: An enhanced AppArmor Docker security profile generator	2019	*Science-Direct*	[55]
A46	Integrity Protection Against Insiders in Microservice-Based Infrastructures: From Threats to a Security Framework	2018	*Springer-Link*	[56]
A47	A survey on security issues in services communication of Microservices-enabled fog applications	2017	*Wiley*	[57]
A48	Integrating Continuous Security Assessments in Microservices and Cloud Native Applications	2017	*ACM*	[58]
A49	Security Audit of Docker Container Images in Cloud Architecture	2021	*IEEE*	[59]
A50	Low-Level Exploitation Mitigation by Diverse Microservices	2017	*Springer-Link*	[60]

**Table 4 sensors-23-01755-t004:** Categorized defense mechanisms and corresponding primary studies.

Potential Defenses of Cloud-Native	
**Defenses**	**References**
Moving Target Defense	A24, A20
Container Security Issues and Their Defenses	A13, A27, A30, A19, A45
Security Attack-Based Protections	A5, A9, A16, A38, A40, A43, A50
Defense-in-Depth	A42
Framework/Architecture-based Solutions	A28, A39

**Table 5 sensors-23-01755-t005:** Identified attacks/vulnerabilities and corresponding primary studies.

Attacks/Vulnerabilities Addressed in the Literature	
**Attacks**	**References**
Server-Oriented Attacks	A21
Injection Vulnerabilities	A25
Container Attacks	A26, A34, A32
Infrastructure and Architectural Attacks	A37, A18
Denial of Service (DoS) Attacks	A41
Integrity Attacks in the MSA	A46

**Table 6 sensors-23-01755-t006:** Identified approaches and tools and corresponding primary studies.

Approaches/Tools in Cloud-Native Systems	
**Approaches/Tools**	**References**
Security Design	A1, A2, A6, A11
Container Tool	A3, A33, A49
Tool Considering Security Services	A8
Detection Mechanism	A22
Risk Analysis	A29
Blockchain	A18, A35
Machine Learning	A38, A40, A34, A41, A26

**Table 7 sensors-23-01755-t007:** Challenges in existing literature and corresponding primary studies.

Challenges in Existing Literature in Cloud-Native Systems	
**Categorization Type**	**References**
Challenges Related to Containers	A15, A21, A4, A17
Challenges Related to Edge and Fog Computing	A12, A47
Systematic Literature Review	A23, A35, A36
Challenges for Practitioners	A7, A14, A44
Challenges in System Design	A10, A31, A48

**Table 8 sensors-23-01755-t008:** Grey literature result in cloud-native systems.

Grey Literature Result in Cloud-Native Systems	
**Categorization Type**	**References**
Tools and Approaches in Industry	[64,65,66,67,68,69,70]
Security Guidance	[71,72,73,74]
Best Practices for Cloud-Native Applications	[75,76,77,78]
Addressing Threats	[79]

## Data Availability

Data available in a publicly accessible repository.

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
