# Peer review of "Static-Analysis-Based Solutions to Security Challenges in Cloud-Native Systems: Systematic Mapping Study"

_sensors, 2023, doi:10.3390/s23041755_

Round 1

Reviewer 1 Report

General comment:

Cybersecurity is a critical concern in today’s digital world, and static analysis-based solutions offer a powerful tool to help protect against a range of threats. Static analysis is a technique used to identify security weaknesses and vulnerabilities in software code without actually running the code. By examining the source code and its structure, static analysis can uncover issues such as memory corruption, buffer overflows, and other security flaws. This type of solution offers several benefits over traditional methods, including improved accuracy and efficiency, a reduced need for manual code reviews, and the ability to detect a greater range of security problems. In this article, we will explore the benefits of static analysis-based solutions to the ever-growing cybersecurity challenges faced by organizations today.

Abstract:

·         Should be organized simple and precise, try to add a general goal/objective and specific goals/objectives…There are three phrases which starts with This study (Line 5), This manuscript (Line 7) and This study (Line 10). Reading these three lines is like you have three different missions.

·         Somehow Line 5 and Line 7 it repeats the same idea. Try to workaround it.

Page 11

-          Line 374. After reading paper [51], the prototype is very interesting but in real practice I do have some doubts about its applicability. As a small suggestion, as per the idea of the article – which is very interesting and I like it very much, there is gap between academic research and industry. Your idea is trying to come and fill somehow a portion of that gap in terms of literature review. Is better also to look on the industry side as well, such as https://www.enisa.europa.eu/publications/enisa-threat-landscape-2022 when it comes about DoS.

 The review covers most of the references that are mostly academic.

 What about the optimization of the static analysis-based solution?

In order to realize the full benefits of static analysis, organizations must optimize their use of the solution. This can be achieved in several ways, including: - Choosing a solution that is best-suited to your organization’s specific needs - Establishing clear and effective communication with the vendor - Taking advantage of the training and educational resources available - Using best practices to optimize the use of the solution - Choosing an integrated solution that can be seamlessly integrated into existing practices and processes

Author Response

Reviewer#1, Concern # 1: Abstract Should be organized simple and precise, try to add a general goal/objective and specific goals/objectives…There are three phrases which starts with This study (Line 5), This manuscript (Line 7) and This study (Line 10). Reading these three lines is like you have three different missions.

Somehow Line 5 and Line 7 it repeats the same idea. Try to workaround it. 

Author response: Thank you for your comment. We agree that organization, precision and simplicity is a key and improved the abstract.

Reviewer#1, Concern # 2: Page 11 Line 374. After reading paper [51], the prototype is very interesting but in real practice I do have some doubts about its applicability. As a small suggestion, as per the idea of the article – which is very interesting and I like it very much, there is gap between academic research and industry. Your idea is trying to come and fill somehow a portion of that gap in terms of literature review. Is better also to look on the industry side as well, such as https://www.enisa.europa.eu/publications/enisa-threat-landscape-2022 when it comes about DoS.

Author response:  Thank you for your observation and suggestion. We agree on the importance of including current industry practices, which can provide us with a comprehensive analysis. We have updated our manuscript by integrating the industry practices by doing a thorough investigation. We considered the suggestions and concentrated on illustrating a bridge between academics and industry in our Result section. We investigate the link and feel the necessity to include this as potential reference. We performed a grey literature review and demonstrated our results in Sub-Section “4.6. Result Extraction of Grey Literature” Line 582. We also discussed the possible best practices practitioners could acknowledge in our Discussion section in Line 721.

Reviewer#1, Concern # 3: What about the optimization of the static analysis-based solution? In order to realize the full benefits of static analysis, organizations must optimize their use of the solution. This can be achieved in several ways, including: 

- Choosing a solution that is best-suited to your organization’s specific needs 

- Establishing clear and effective communication with the vendor 

- Taking advantage of the training and educational resources available 

- Using best practices to optimize the use of the solution 

- Choosing an integrated solution that can be seamlessly integrated into existing practices and processes

Author response:  

Thank you for thorough analysis and comments. Addressing to your comments:

 - Choosing a solution that is best-suited to your organization’s specific needs

 - response: From our findings, we have classified the potential defenses in cloud-native applications in the sub-section “4.2. Security Defense Mechanism in Cloud-Native Systems: RQ1”. In that classification, we broadly described the security-based solutions and defined in which situation we can utilize those solutions from Line 352 to 360.

 -  Establishing clear and effective communication with the vendor

 - response:  The effective needs to establish the communication with the vendor are missing in the primary studies. We tried to construct a overview to establish a security defense mechanism which the practitioner can utilize following the current best practices and the mindmap we have developed.

 - Taking advantage of the training and educational resources available

- response: We have yet to find any material for academic research to train the teams about the defensive mechanism. However, the best practices can be potentially utilized, which we addressed in our Result and Discussion Section.

 - Using best practices to optimize the use of the solution

- response: We significantly categorized the grey literature where we classified the studies broadly described the industry best practices in Result Section Line 591. In addition, we have considered those assumptions and practices and made a separate consideration in our Discussion Section, Line 731.

- Choosing an integrated solution that can be seamlessly integrated into existing practices and processes

- response: We comprehensively investigated the security solutions based on static analysis. The solutions primarily focused on the protection of the APIs and access control mechanism to ensure confidentiality. However, the integrated solution which can be integrated into the existing processes are not discovered in our finindings. Commonly, the largely scalability of microservice solutions and the practice security by design can comprehensively address this.

Reviewer 2 Report

Summary: This study aims to present a thorough overview of the security defensive mechanisms now in use that can identify attacks and respond to those assaults and vulnerabilities. Additionally, it tries to draw attention to future research needs. In order to characterize static analysis-based protection mechanisms to thwart security assaults in cloud-native systems leveraging microservices, this research does a thorough literature survey. Additionally, it isolates the existing methods for dealing with security analysis of microservices and security principle violations.

Comments and Suggestions:

- The paper is well-written and well-structured. In addition, it covers an interesting topic.

-  In the title of the paper the expression "mapping study" should be capitalized.

- The authors may add a new figure that illustrates the structure of the paper.

- The sections are unbalanced in length. The authors need to solve this issue. 

- The authors may also include more figures and tables that summarize the content presented in some sections of the paper for the purpose of helping readers reach and memorize important results quickly and efficiently.

- The use of blockchain technology in the considered context needs to be considered in more detail.

- Similarly, the use of Artificial Intelligence for the same purpose has to be covered. In this direction, the authors are invited to consider the following reference (and others): https://dl.acm.org/doi/10.1145/3433174.3433614

- Also the authors need to report on the industrial techniques and methodologies adopted for guaranteeing security for the considered types of systems.

Author Response

Reviewer#2, Concern # 1:  The paper is well-written and well-structured. In addition, it covers an interesting topic.

Author response: Thank you for your comments and feedback.

Reviewer#2, Concern # 2:  In the title of the paper the expression "mapping study" should be capitalized.

Author response:  Thank you for this comment and thorough catch. We have updated this in our manuscript.

Reviewer#2, Concern # 3:  The authors may add a new figure that illustrates the structure of the paper.

Author response: Thank you for your comment. We have analyzed other similar studies about this concern and did not find a suitable figure for the structure of the paper, other than what we shared in Figure 8 as a categorization outcome, or Figure 1 details the selection methodology.

Reviewer#2, Concern # 4:  The sections are unbalanced in length. The authors need to solve this issue. 

Author response: Thank you for your comments and feedback. We have tried to manage the sections with a more in-depth analysis of our findings. In addition, we have rewritten the discussion sections with more content based on the reviews we got from our respected reviewers.

Reviewer#2, Concern # 5:  The authors may also include more figures and tables that summarize the content presented in some sections of the paper for the purpose of helping readers reach and memorize important results quickly and efficiently.

Author response: We appreciate this comment, and we followed your suggestions. In addition, we have included several figures for our results analysis for each classification. Figure 3 on Page 9, Figure 4 on Page 11, Figure 5 on Page 13, Figure 6 on Page 15, Figure 7 on Page 17 are depicted to address the feedback. 

Reviewer#2, Concern # 6:  The use of blockchain technology in the considered context needs to be considered in more detail.

Author response: Thank you for your suggestion and feedback. We agree on the point of giving the topic a more in-depth analysis. For this reason, in the Result Sub-Section “4.4. Tools or Strategies in Security Defense Mechanism: RQ3” Line 467, we have separated the analysis. In addition, in our Discussion section Line 756, we have concluded our investigation and recommend how this mechanism can be utilized in the security analysis of cloud-native.

Reviewer#2, Concern # 7:  Similarly, the use of Artificial Intelligence for the same purpose has to be covered. In this direction, the authors are invited to consider the following reference (and others): https://dl.acm.org/doi/10.1145/3433174.3433614

Author response: Thank you for your thorough analysis and feedback. We considered the importance of including Artificial Intelligence and included it as a particular point in the Result Sub-Section “4.4. Tools or Strategies in Security Defense Mechanism: RQ3” in Line 478. In addition, we similarly include the integration of AI in our Discussion section in line 767. Finally, we investigated the article provided and felt the need to have that as our potential study.

Reviewer#2, Concern # 8:  - Also the authors need to report on the industrial techniques and methodologies adopted for guaranteeing security for the considered types of systems.

Author response: Thank you for your feedback and suggestions. We acknowledge the value of considering current industry practices since they can provide us with a thorough examination. We extensively examined the industry standards and modified our document to reflect them. We took the advice into account and focused on showing how academia and industry can coexist in our Result section. Finally, we examined the grey literature and Sub-Section "4.6. Result Extraction of Grey Literature" Line 582 demonstrates our findings. Furthermore, we spoke about potential best practices practitioners may adopt in Line 721 of our Discussion section.

Round 2

Reviewer 2 Report

The authors took into consideration my remarks and suggestions. I have no other comments to make. Good luck.